# Pumping up the Fight against Multiple Sclerosis: The Effects of High-Intensity Resistance Training on Functional Capacity, Muscle Mass, and Axonal Damage

**DOI:** 10.3390/healthcare12080837

**Published:** 2024-04-15

**Authors:** Sergio Maroto-Izquierdo, Patricia Mulero, Héctor Menéndez, José Pinto-Fraga, Simone Lista, Alejandro Santos-Lozano, Nieves Téllez

**Affiliations:** 1i+HeALTH, Strategic Research Group, Department of Health Sciences, Miguel de Cervantes European University (UEMC), 47012 Valladolid, Spain; 2Neurology Department, Hospital Clínico Universitario de Valladolid, 47003 Valladolid, Spain

**Keywords:** neurofilaments, neurodegeneration, exercise, strength, hypertrophy, physical function, multiple sclerosis

## Abstract

Background: Resistance training (RT) has been recognized as a beneficial non-pharmacological intervention for multiple sclerosis (MS) patients, but its impact on neurodegeneration is not fully understood. This study aimed to investigate the effects of high-intensity RT on muscle mass, strength, functional capacity, and axonal damage in MS patients. Methods: Eleven relapsing–remitting MS patients volunteered in this within-subject counterbalanced intervention study. Serum neurofilament light-chain (NfL) concentration, vastus lateralis thickness (VL), timed up-and-go test (TUG), sit-to-stand test (60STS), and maximal voluntary isometric contraction (MVIC) were measured before and after intervention. Participants performed 18 sessions of high-intensity RT (70–80% 1-RM) over 6 weeks. Results: Significant (*p* < 0.05) differences were observed post-intervention for VL (ES = 2.15), TUG (ES = 1.98), 60STS (ES = 1.70), MVIC (ES = 1.78), and NfL (ES = 1.43). Although moderate correlations between changes in VL (R = 0.434), TUG (R = −0.536), and MVIC (R = 0.477) and changes in NfL were observed, only the correlation between VL and MVIC changes was significant (R = 0.684, *p* = 0.029). Conclusions: A 6-week RT program significantly increased muscle mass, functional capacity, and neuromuscular function while also decreasing serum NfL in MS patients. These results suggest the effectiveness of RT as a non-pharmacological approach to mitigate neurodegeneration while improving functional capacity in MS patients.

## 1. Introduction

Multiple sclerosis (MS) is a chronic autoimmune disorder of the central nervous system that affects over 2.5 million people worldwide [1]. It is characterized by inflammation, demyelination, and axonal damage, leading to irreversible neurological disability. Symptoms include cognitive impairment, sensory and motor deficits, spasticity, fatigue, and impaired balance and coordination [2], often resulting in decreased functional capacity, reduced quality of life, and an increased risk of falls and injuries [3].

Despite the availability of pharmacological therapies for MS, exercise has emerged as a non-pharmacological approach that improves both physical and psychological health [2]. In particular, resistance training (RT) has been shown to effectively enhance muscle mass, functional capacity, and quality of life in MS patients [4]. Previous studies have demonstrated that RT can increase muscle strength [3,5], improve balance and gait [4], reduce fatigue [6], and even promote neuroplasticity [7] in MS patients. Additionally, the benefits of RT on strength and functionality have been shown to surpass those of traditional MS care strategies, such as moderate aerobic exercise [3]. However, its effects on neuroinflammation and neurodegeneration are not yet fully understood.

Neurofilaments (NFs) are neuronal cytoskeletal proteins made up of NFs light (NfL), NFs medium (NfM), NFs heavy (NfH) chains, and alpha-internexin located specifically in the neuro-axonal compartments [8]. These proteins support neuron cytoskeletons and are released into the interstitial fluid and subsequently into both cerebrospinal fluid (CSF) and the bloodstream following axonal membrane damage [9]. Elevated NF concentrations have been linked to increased disease activity and disability in MS patients [10]. NfL, due to its small molecular weight and high solubility, effectively diffuses from the parenchyma into the CSF after axonal degeneration occurs [11]. Changes in NfL concentrations in biofluids are associated with brain atrophy and injury, as evidenced in mouse models and various human neurological disorders [12,13,14,15]. Particularly, recent studies have explored blood (plasma/serum) NfL concentration as a potential biomarker of neurodegeneration in MS [8,16].

Exercise is known to have neuroprotective effects in both animal models and humans [17]. Notably, RT has been found to increase concentrations of brain-derived neurotrophic factor (BDNF), a growth factor that promotes neuronal survival and plasticity [18,19]. This suggests that greater increases in muscle strength are associated with greater release of neurotrophic factors like BDNF [3,18,20]. Recently, we investigated the effects of high-intensity RT (i.e., intensity above 70% of 1-RM) on neurodegeneration in MS patients [21]. Our study documented a significant decline in serum NfL concentrations after a six-week period of RT, contributing to the growing evidence of its efficacy. Such training is increasingly recognized as a viable, safe, and well-tolerated exercise form, potentially superior to moderate (60–70% of 1-RM) or low-intensity (<60% of 1-RM) alternatives. Showing promise in enhancing muscle size and quality, augmenting muscle strength and power, boosting neural drive, improving functional ability and cognitive function, and reducing disease severity and self-reported fatigue in MS patients [22,23,24,25,26,27,28,29]. Furthermore, organizing the prescribed exercises in a circuit format may have significant effects on cardiorespiratory function and body composition [30], potentially surpassing those achieved with moderate-intensity aerobic exercise [31].

Despite these advances, it remains uncertain whether the observed decrease in NfL concentrations is directly correlated with improvements in functional and neuromuscular capacities, warranting further investigation into how high-intensity RT may not only improve MS patients’ functional capacity but also potentially slow down neurodegeneration progression. Therefore, this study aimed to investigate the effects of a 6-week high-intensity RT training program on muscle mass, functional capacity, muscle strength, and neurodegeneration assessed using serum NfL concentrations in MS patients. We hypothesize that high-intensity RT induces significant changes in functional capacity, muscle mass, and strength, correlating positively with a reduction in serum NfL concentrations. Increases in neuromuscular capacity could, therefore, be positively correlated with decreases in axonal damage.

## 2. Materials and Methods

### 2.1. Study Design

We conducted a longitudinal, within-subject, counterbalanced pilot study. MS patients attended the laboratory on three occasions (i.e., tests 1, 2, and 3; see Figure 1). At each visit, we measured serum NfL, vastus lateralis (VL) thickness, timed up-and-go test (TUG), 60 s sit-to-stand test (60STS), and maximal voluntary isometric contraction (MVIC) during squat and leg extension exercises. Additionally, anthropometric measurements (height, weight, and body fat percentage via bioimpedance) were taken during the first laboratory visit. The first visit (test 1) was performed seven weeks before the second laboratory visit (test 2). Between test 1 and test 2, participants were asked to avoid intense exercise and continue their usual care routines. After test 2, participants were familiarized with the 6 strength exercises prescribed. Individual exercise intensities were calculated according to the American College of Sports Medicine (ACSM) protocol for estimating the one-repetition maximum (1-RM). This estimation is based on the number of repetitions performed to fatigue with a submaximal load and was calculated using the average value from the formulas of Brzycki, Epley, Lander, Cummings, O’Conner, and Abadie. One week later, MS patients completed 18 sessions (3 sessions per week for 6 weeks) of high-intensity circuit-based RT (70–80% of 1-RM) under supervision. Final assessments (test 3) were conducted one week after completing the last training session. Conducted in accordance with the Helsinki Declaration to ensure ethical research practices, this study was approved by the institutional Ethical Committee (PI 21-2399), with all participants providing written informed consent.

### 2.2. Participants

Eleven patients with relapsing–remitting MS voluntarily participated in this study. Participants were eligible for inclusion if they (i) had a diagnosis of relapsing–remitting multiple sclerosis in accordance with the 2017 International Panel Diagnostic Criteria, (ii) experienced no relapse or change in MS treatment in the six months prior to test 1, (iii) showed no new demyelinating lesions in T2 sequences or gadolinium-enhancing lesions on a brain magnetic resonance imaging (MRI) performed within 12 months before test 1, and (iv) had a standardized Expanded Disability Status Scale (EDSS) score of ≤3 (please refer to Table 1 for the characteristics of the included participants). Participants reporting a high level of physical activity (i.e., engaging in vigorous physical activity for more than three days per week) at baseline, as measured by the International Physical Activity Questionnaire, or those previously involved in any RT program, were excluded. All participants successfully completed all aspects of the study, which included two familiarization sessions, 18 supervised training sessions, and baseline, pre-, and post-training tests.

Additionally, we calculated the statistical power post hoc to assess the likelihood of detecting a significant effect, given the observed effect size, sample size, and the alpha level set for our analyses. The statistical power calculation was performed using the G*Power software (G*Power 3.1.9.2, Heinrich-Heine-Universitat Dusseldorf, Dusseldorf, Germany; http://www.gpower.hhu.de/, accessed on 22 January 2024). For this calculation, we inputted an expected effect size of 0.5, consistent with medium effect sizes as outlined by Cohen [32] and previous studies [3], and an alpha level of 0.05, reflecting the standard threshold for statistical significance in biomedical sciences research. Given the exploratory nature of our study and the specific challenges associated with recruiting participants from a narrowly defined population of patients with MS, our total sample comprised 11 subjects. This sample size, along with the allocation of participants across three measurement points within our study design, was factored into our power analysis. The result of this computation revealed a statistical power of 0.22.

### 2.3. Procedures

#### 2.3.1. NfL Measurement

Venous blood samples (10 mL) were collected from the medial cubital vein into EDTA tubes at three different time points: baseline (test 1), one week before training (test 2), and one week after completing the training program (test 3). For each sample, the serum fraction was collected, aliquoted into multiple 0.5 mL cryovial-sterilized tubes, and then stored at −80 °C until analyzed at the Immunology Department of Hospital Ramón y Cajal (Madrid, Spain). Serum NfL concentrations were quantified using single-molecule array technology (SiMoA; Quanterix Corp) with a SR-X instrument (Quanterix, Lexington, MA, USA) and the SiMoA human NF-light Advantage Kit (Quanterix, Billerica, MA, USA). A detailed protocol and data specific to serum NfL are comprehensively described in one of our prior studies [21].

#### 2.3.2. Muscle Thickness

Muscle size for the vastus lateralis (VL) was measured 30 min after blood collection. The protocol previously described by Maroto-Izquierdo and colleagues [33] was used to measure the thickness of the VL muscle of the dominant leg in all participants. B-mode ultrasound (Esaote, MyLab Gamma, Esaote Spa, Genoa, Italy) was used to measure the VL muscle in vivo with a 6 cm linear array transducer at 12 MHz, which was coated with a water-soluble transmission gel for acoustic contact without depressing the dermal surface. Participants were placed on a bed for 10–15 min before the measurements to allow osmotic fluids to stabilize. Muscle thickness was measured at 50% distance between the greater trochanter and lateral condyle of the femur, with the ultrasound transducer placed in the transverse plane and perpendicular to the skin. The thickness of the VL muscle was determined by measuring the perpendicular distance between the superficial and deep aponeurosis of the muscle before and after intervention using ultrasound software. To enhance data reliability, measurement points were marked on the skin with indelible ink daily, and muscle architecture was compared pre- and post-intervention. Each measurement was conducted three times, with the average of these measurements used to ensure result reliability.

#### 2.3.3. Functional Capacity

Functional capacity was assessed through the TUG and 60STS tests. The TUG test, conducted according to the method described by Karlon et al. [34], began with the participant seated on a standard-height chair (45 cm), back touching the backrest, and arms resting on the armrests. Participants were instructed to stand, walk 3 m, turn around, return to the chair, and sit. The timer started as they began to stand and stopped once they sat back down. A decrease in the TUG test time indicates an improvement in walking ability. Participants wore their regular footwear and used any usual assistive devices. One minute after the TUG test, the 60STS test was performed following Bohannon’s guidelines [35], using a standard 45 cm chair without armrests. Participants were to sit with feet flat on the ground, then stand fully, extending knees and hips, and sit again with arms crossed over their chest, repeating as quickly as possible for 60 s. The total number of completed STS repetitions was recorded. The 60STS test, being maximal, was conducted only once.

#### 2.3.4. Neuromuscular Capacity

Five min after the 60STS test, we measured muscle force during the squat (MVIC_SQ_) and leg extension exercises (MVIC_LEXT_). For the isometric squat, a dual-force plate system (Hawkin Dynamics, Westbrook, ME, USA) was used. We positioned the patients on the platforms with their knees flexed at 100°. A harness was used to attach participants to the platform system through a cable, allowing individualized height adjustment for each patient. Bilateral peak force (MVIC_SQ_), relative force (RF, N/kg), peak force of each leg, as well as rate of force development (RFD) from 0 to 100 ms and from 0 to 250 ms were recorded. For the leg extension, participants sat on a locked leg extension device with a portable strain gauge with its own software (Chronojump, Barcelona, Spain) sampling at 80 Hz secured at the end of the device. The other end was connected to a chain linked to a padded anklet, designed specifically for maximal isometric testing to ensure mechanical rigidity and minimize joint movement [36]. The chain length was adjusted to each participant’s anthropometric characteristics to achieve mechanical rigidity with a comfortable knee joint angle of 90°. Before each trial, knee flexion was measured using a handled goniometer (Nexgen Ergonomics, Point Claire, QC, Canada). The leg extension exercise was performed bilaterally and also unilaterally with both affected (A) and non-affected limbs (NA). For both exercises, patients were asked to perform a MVIC as fast as possible and to maintain muscle contraction for 5 s. Two attempts were allowed for each test with a 2 min rest between trials. Only the repetition with the highest peak force was used for subsequent analysis.

#### 2.3.5. Training Intervention

The intervention consisted of 18 high-intensity, circuit-based RT sessions over 6 weeks (3 sessions per week), with 48 h of rest between sessions. Each session included three sets of eight to ten repetitions with coupled concentric and eccentric muscle actions of seven exercises targeting all major muscle groups, following recent scientific recommendations for moderate to high intensities and volumes [3]. The intensity of the exercises progressively increased from 70% to 80% of the one-repetition maximum, as outlined in Table 2, with two minutes of rest between circuit sets but no rest between exercises. The individual exercise intensity was determined using the ACSM protocol for estimating the 1-RM one week before starting the intervention. A 5 min moderate-intensity warm-up on an elliptical machine preceded each session. Qualified exercise professionals with degrees in Sport Sciences and experience in strength training for special populations supervised all sessions, providing verbal cues and assistance to participants.

### 2.4. Statistical Analysis

Normality was checked by the Shapiro–Wilk normality test. Then, a two-factor (condition; time) repeated-measures analysis of variance (ANOVA) and Bonferroni post hoc tests were used to investigate differences in variables after intervention within participants and between groups. The effect size (ES) was calculated for interactions between groups using Cohen’s guidelines. Threshold values for ES were >0.2 (small), >0.6 (large), and >2.0 (very large) [32]. The Pearson correlation coefficient was utilized to assess the relationship between changes in NfL and changes in each outcome variable. A correlation coefficient value less than 0.1 indicated negligible correlation, between 0.1 and 0.39 indicated weak correlation, between 0.4 and 0.69 indicated moderate correlation, between 0.7 and 0.89 indicated strong correlation, and any R value above 0.9 indicated very strong correlation [37]. Additionally, linear regression analyses were performed to determine the predictive equations, with functional or structural changes as independent variables and NfL changes as the dependent variable.

All statistical analyses were performed using the Jamovi software package (The Jamovi Project, v.1.6.23.0; downloadable at https://www.jamovi.org, accessed on 10 April 2024). The level of significance for all tests was set to *p* < 0.05. Mean, standard error (SE), and the t value were reported for all statistical analyses.

## 3. Results

All data (mean and SD) for each of the variables by endpoints are shown in Table 3. In addition, the results of the ANOVA for the magnitude of increase after each condition showed a significant difference in mean scores in VL (19.1, *p* = 0.002, t_(10)_ = 4.1, ES = 1.23), TUG (−12.41, *p* < 0.001, t_(10)_ = −4.4, ES = 1.32), MVIC_LEXT_ (25.5, *p* = 0.004, t_(10)_ = 3.7, ES = 1.12), MVIC_SQ_ (49.0, *p* = 0.004, t_(10)_ = 3.8, ES = 1.20), RF in the MVIC_SQ_ (38.4, *p* < 0.001, t_(10)_ = 4.7, ES = 1.49), and NfL (39.6, *p* = 0.005, t_(10)_ = 3.5, ES = 1.07) between the experimental and control conditions (Figure 2).

The ES was large for these outcomes, indicating a substantial difference between the two conditions. However, no significant correlation was observed between the decrease in serum NFL concentration and the increases in muscle mass, functional capacity, or neuromuscular capacity, although changes in VL (R = 0.434, *p* = 0.182), TUG (R = −0.536, *p* = 0.089), and MVIC_SQ_ (R = 0.477, *p* = 0.163) showed a moderate correlation with NfL changes. Additionally, changes in VL and MVIC_SQ_ showed a significant, moderate correlation (R = 0.684, *p* = 0.029). Finally, no significant differences were observed between the experimental and control conditions for 60STS, unilateral (A and NA) MVIC_SQ_, RFD 0-100, and RFD 0-250.

## 4. Discussion

This study aimed to explore the effects of a short-term, high-intensity RT program on serum NfL concentrations, muscle mass, functional capacity, and neuromuscular function in mildly disabled MS patients. The findings revealed that a 6-week RT program significantly increased muscle mass, functional capacity, and neuromuscular function while also decreasing serum NfL concentrations. These results underscore the potential of high-intensity RT as an effective non-pharmacological co-adjuvant intervention for reducing neuroinflammation and neurodegeneration and improving functional capacity in MS patients, though more research is needed to ascertain the extent and persistence of these protective effects. Furthermore, these results prompt us to propose the hypothesis that enhancements in muscle growth and muscle strength might serve as indicators of neuronal preservation. While this concept is plausible, it necessitates further investigation to confirm its validity.

In recent years, exercise has been recommended as the primary non-pharmacological intervention for treating patients with MS [38]. Exercise-based interventions have been shown to alleviate MS symptoms, including cognition and sphincter function, fatigue, and overall quality of life [3]. However, the underlying pathophysiological mechanisms of this phenomenon are not yet fully understood. Regular exercise training was suggested by studies in both animal models and MS patients to stimulate the secretion of neurotrophic factors [39], modulate the immune system [40], improve cytokine balance [41], decrease the interferon response [42], and promote the mobilization and recruitment of neural progenitor cells at lesion sites [43]. The reported reduction in serum NfL concentrations suggests a potentially protective effect of RT against axonal loss. NfL is a structural protein of the axon cytoskeleton, gaining increasing evidence as a candidate biomarker for tracking axonal loss [16]. Elevated serum NfL concentrations were found to be associated with clinical and image outcomes in MS and other neurodegenerative diseases, including a higher relapse rate, disability, T2 lesion load in MRI, and brain atrophy [21].

In this line, disease-modifying treatments were shown to decrease NfL concentrations as a marker of axonal preservation [16]. Previous research reported that a single session of high-intensity aerobic interval exercise [10] or eight weeks of moderate aerobic exercise [44] also led to reduced NfL concentrations in MS patients. The present study demonstrated that a short-term high-intensity RT program can also reduce NfL concentrations in mildly disabled MS patients while improving muscle mass and functional and neuromuscular capacity. Therefore, higher neuromuscular adaptations induced by resistance exercise training are expected to potentially lead to greater and longer-lasting protective effects against neuroinflammation and neurodegeneration. However, this hypothesis requires further investigation to be substantiated. Indeed, future analyses are needed to explore the time-course and length of this phenomenon, as well as to make comparisons between different exercise protocols (i.e., aerobic vs. RT).

While no significant correlation was observed between the decrease in serum NfL concentrations and the increase in muscle thickness after training, high-intensity RT significantly increased muscle mass. Increasing muscle mass in MS patients could have neuroprotective effects, as muscle fibers produce neurotrophins such as BDNF, insulin-like growth factor 1 (IGF-1), and vascular endothelial growth factor (VEGF) [45] that support nerve cell growth and survival. Increased muscle mass can also enhance physical function and improve mobility and balance, thus improving overall quality of life. It may prevent sarcopenia and muscle weakness [46], common in MS, which impair mobility and daily activities [47]. Additionally, maintaining a healthy weight is crucial for overall health and disease management. Although previous studies also observed increases in lean mass following a high-intensity RT program in MS patients [29], this study is unique in demonstrating muscle thickness increases through ultrasound in this population. The underlying physiological mechanism for muscle mass increase remains speculative but may involve muscle remodeling through growth factors and myokines, myoblast proliferation activation, or changes in muscle proteome after resistance exercise, enhancing the anabolic environment [25].

Similarly, we found a non-significant moderate correlation between the increases in force production capacity and the decrease in serum NfL values following the training program. However, significant correlations were observed between the increases in muscle mass and isometric strength improvements after the intervention. Therefore, the strength gains, underscored by our results and those from previous studies on high-intensity RT in MS patients [3], may be partly attributed to increased muscle mass. Prior analyses have reported isometric strength gains of 7–18% in knee extensors after moderate-intensity RT programs in MS patients [5,27,48,49]. Our study showed a 22% increase in MVIC for knee extension (with approximately 12% of increases in each leg). These results are slightly higher than those observed in previous studies, which may be due to the higher training intensity (70–80% of 1-RM) used compared to similar studies [3]. We uniquely assessed maximum strength bilaterally and unilaterally during the 100°-knee flexion squat, noting significant increases post-training. Despite similar improvements across affected and unaffected limbs, RFD measurements during the squat did not significantly change, diverging from observations in the general healthy population [50]. This discrepancy might stem from MS’s impact on the central nervous system [51,52], affecting rapid force generation and leading to neuromuscular dysfunction and efficiency losses at the neuromuscular junction [53]. Thus, muscle strength impairments in MS are most pronounced during maximal dynamic contractions [53]. The characteristic fatigue and muscle weakness in MS may also hinder rapid force generation [53,54], with notable inter-subject variability in training responses, suggesting the need for larger samples to fully understand changes in neuromuscular capacity as measured by RFD.

These increases in muscle strength were associated with positive changes in functional capacity. Previous studies reported significant improvements in the TUG test (~22%) [28,48,55], which are similar to the results of our study. Likewise, the 60STS revealed significant improvements only after the training program, with approximately 20% improvements, like those shown in previous studies that prescribed high-intensity RT in patients with MS (i.e., 60–85% of 1-RM) [49,55], considered clinically relevant [56]. To date, these changes have been linked to an increased capacity to generate force [3]. Interestingly, we observed reductions in serum NfL concentrations alongside improvements in functional capacity in MS patients. However, establishing a causal relationship between these functional improvements and neurodegenerative biomarkers, such as NfL, requires further investigation. This is crucial to confirm the utility of functional assessments as reliable indicators of neurodegenerative progression in a clinical setting.

Furthermore, this pilot study has demonstrated the feasibility of prescribing high-intensity circuit-based RT for individuals with MS. Circuit-based RT is a method of organizing a RT session that optimizes time and generates a greater metabolic stimulus [30]. It has been shown to improve not only strength but also body composition and cardiorespiratory capacity to a greater extent than traditional training in healthy adults [30]. However, high-intensity circuit-based RT is not a methodology widely employed in studies with MS patients. In our study, this prescription did not affect adherence or neuromuscular performance. While the impact of this intervention on cognitive and cardiorespiratory parameters remains unknown, engaging MS patients in high-intensity interval exercise has been associated with greater increases in maximal oxygen consumption and cognitive capacity [31]. Future research should analyze the effect of circuit-based RT on this population, focusing on process and behavioral, cognitive, and cardiorespiratory variables.

We recognize that a primary limitation in our study stems from the absence of a blinded control group. Despite this, the validity of our findings is supported by the observed stability in serum NfL concentrations over a brief period, coupled with the meticulous selection of patients who were clinically and radiologically stable. Another critical constraint was our sample size. Given that previous research documented reductions in NfL concentration within the initial three months following the commencement of new treatments [57], we restricted our participant pool to MS patients consistently undergoing the same treatment regimen for a minimum of six months. This selection criterion significantly narrowed our sample size, despite being comparable to those of similar previous studies [54,58,59].

Thus, the study’s relatively low statistical power underscores the preliminary nature of our findings and the challenges inherent in researching special populations like MS. With a power level below the preferred 0.80 threshold, we highlight the importance of cautious interpretation and view our work as a foundational step for future research. This study signals the need for larger sample sizes and more robust methodologies in upcoming investigations to enhance statistical rigor and address the limitations that currently hinder our ability to detect significant effects and relationships. Such constraints notably increase the risk of Type I errors and limit the generalizability of our conclusions. Moving forward, it is warranted that future research, particularly studies involving randomized interventions, recruits larger cohorts. This approach will not only solidify our findings but also significantly contribute to a more nuanced understanding and evaluation of both clinical and radiological outcomes in MS.

## 5. Conclusions

A 6-week high-intensity RT program significantly decreased serum NfL concentrations while also improving muscle thickness, lower limb maximal voluntary isometric contraction, and performance in the timed up-and-go and sit-to-stand tests in MS patients. Although gains in muscle thickness and isometric strength showed a significant correlation, no significant correlation was observed between increases in muscle mass and strength and the decrease in NfL concentrations. In addition, the magnitude of the changes experienced after the training program was significantly greater than those reported after the control condition, in which no exercise was performed. These findings emphasize the potential usefulness of resistance exercise training (conducted three times per week and involving at least one exercise from each major muscle group at an intensity above 70% of the 1-RM) as an effective non-pharmacological intervention to reduce axonal damage and neurodegeneration while improving functional capacity and muscle quality in MS patients. Future studies should explore the comparative effectiveness of different types of strength exercises and training modes on axonal damage to identify optimal training protocols that maximize benefits for MS patients, addressing a critical gap in current research and offering a more tailored approach to managing MS symptoms.

## Figures and Tables

**Figure 1 healthcare-12-00837-f001:**
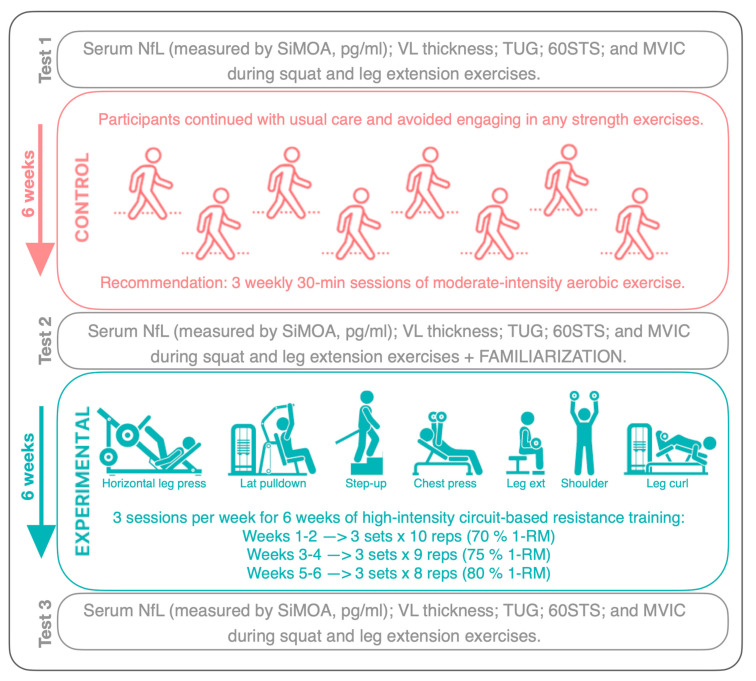
Schematic representation of the study design.

**Figure 2 healthcare-12-00837-f002:**
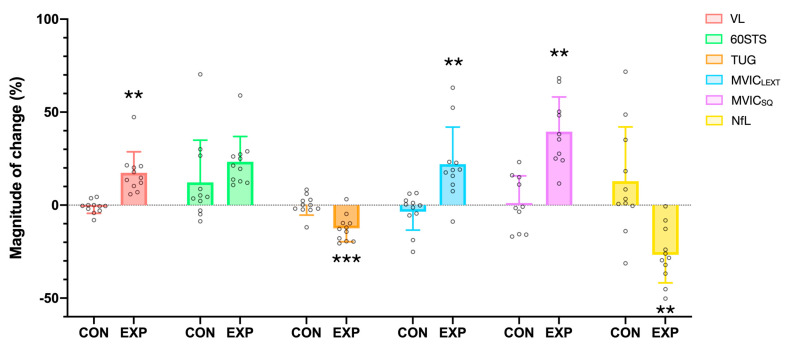
Relative changes from the baseline values (mean ± SD) in vastus lateralis thickness (VL, red), muscle endurance (XRM, green), time up-and-go (TUG, orange), leg extension MVIC (LExt, blue) and squat MVIC (SQ, purple), and NfL (yellow) in the control (CON) and experimental (EXP) conditions. ** Indicates significant differences from the control group at *p* < 0.01. *** Indicates significant differences from the control group at *p* < 0.001.

**Table 1 healthcare-12-00837-t001:** Demographic and clinical baseline characteristics of participants. Data are expressed as mean ± SD or absolute value (%).

Baseline Characteristics	*n* = 11
**Antropometric data**	
Sex; F/M (% women)	9/2 (81.8%)
Age (years)	40.8 ± 7.8
Weight (kg)	66.1 ± 13.7
Height (cm)	160.5 ± 9.5
Fat percentage (%)	32.2 ± 7.1
**Disease characteristics**	
Relapsing–remitting MS	11 (100%)
Secondary progressive MS	0 (0%)
Primary progressive MS	0 (0%)
Progressive-relapsing MS	0 (0%)
Last relapse	8.3 ± 4.1
Years since diagnosis	12.1 ± 6.7
EDSS	0.5 ± 0.8
**Treatment**	
None	2 (18.2%)
Interferon beta	3 (27.3%)
Dimethyl Fumarate	2 (18.2%)
Teriflunomide	1 (9.1%)
Fingolimod	1 (9.1%)
Cladribine	1 (9.1%)
Alemtuzumab	1 (9.1%)
**Habits**	
Low physical activity level (IPAQ scale)	3 (27.3%)
Moderate physical activity level (IPAQ scale)	8 (72.7%)
Smoking	1 (9.1%)

Abbreviations: EDSS, Expanded Disability Status Scale; IPAQ: International physical activity questionnaire; MS, multiple sclerosis.

**Table 2 healthcare-12-00837-t002:** Exercise intensity and volume dose used in each session for each training week and the exercises prescribed in the resistance circuit-based training.

Weeks	Intensity	Volume	Resistance Exercises
Weeks 1–2	70% 1-RM	3 sets × 10 reps	Horizontal leg press. Lat pulldown. Cable step-up. Chest press machine. Leg extension. Dumbbell shoulder press. Leg curl.
Weeks 3–4	75% 1-RM	3 sets × 9 reps	
Weeks 5–6	80% 1-RM	3 sets × 8 reps	

**Table 3 healthcare-12-00837-t003:** Changes (mean ± SD) in muscle mass, functional capacity, muscle force and rate of force development, and neurodegeneration before control condition (test 1), after control condition (test 2), and after training intervention (test 3), *p* value for the comparison between test values by Bonferroni test, mean differences with 95% confidence interval, and effect size (ES) for the changes.

					Test 2 vs. Test 1				Test 3 vs. Test 1				Test 3 vs. Test 2		
	Test 1	Test 2	Test 3		*p*	Mean (95CI)	ES		*p*	Mean (95CI)	ES		*p*	Mean (95CI)	ES
**Muscle mass**															
VL (mm)	18.6 ± 3.8	18.5 ± 4.1	21.5 ± 3.1		0.506	0.1 (−0.6–0.2)	0.21		<0.001	2.8 (1.9–3.7)	2.22		<0.001	2.9 (2.0–3.9)	2.15
**Functional capacity**															
60STS (n)	39.8 ± 8.9	43.5 ± 6.9	53.2 ± 7.1		0.089	3.6 (−0.7–7.9)	0.57		<0.001	13.4 (7.2–19.6)	1.45		<0.001	9.7 (6.4–13.0)	1.98
TUG (s)	5.4 ± 0.4	5.4 ± 0.5	4.7 ± 0.4		0.953	0.0 (−0.2–0.2)	0.02		<0.001	0.7 (0.4–1.0)	1.55		<0.001	0.69 (0.4–0.9)	1.70
**Muscle force**															
MVIC_LEXT_ (N)	551 ± 263	529 ± 249	647 ± 314		0.163	−22.2 (−55.0–10.6)	0.45		0.017	96.0 (21.5–170.5)	0.87		0.005	118.2 (45.7–190.6)	1.10
MVIC_LEXT_ NA (N)	276 ± 136	285 ± 124	320 ± 159		0.619	9.3 (−31.1–49.7)	0.16		0.022	44.9 (8.0–81.8)	0.82		0.152	35.6 (−15.5–86.7)	0.47
MVIC_LEXT_ A (N)	279 ± 128	266 ± 122	297 ± 140		0.544	−12.7 (−57.7–32.3)	0.19		0.409	18.6 (−29.5–66.6)	0.26		0.267	31.3 (−28.0–90.5)	0.36
MVIC_SQ_ (N)	1809 ± 723	1725 ± 690	2394 ± 630		0.274	84.0 (−247.0–79.1)	0.37		<0.001	585.7 (351.0–820.1)	1.79		<0.001	669.7 (401.0–938.6)	1.78
MVIC_SQ_ NA (N)	829 ± 154	836 ± 137	1186 ± 239		0.889	6.6 (−102.0–115.0)	0.05		0.002	293.1 (145.0–442.0)	1.41		0.006	337.4 (135.0–540.0)	1.39
MVIC_SQ_ A (N)	819 ± 188	736 ± 112	1168 ± 304		0.167	−82.3 (−208.0–44.0)	0.54		0.003	252.6 (114.0–390.9)	1.31		<0.001	386.8 (228.0–545.5)	2.04
RF (N·kg^−1^)	280 ± 87	276 ± 67	381 ± 87		0.745	−4.2 (−32.7–24.2)	0.11		<0.001	100.9 (58.5–143.3)	1.70		<0.001	105.1 (68.1–142.1)	2.03
RFD 0–100 ms (N·s^−1^)	1323 ± 427	1640 ± 445	1790 ± 427		0.483	317.0 (−663.0–1297.0)	0.23		0.172	467.0 (−245.0–1178.0)	0.47		0.710	150.0 (−733.0–1033.0)	0.12
RFD 0–250 ms (N·s^−1^)	1061 ± 236	1111 ± 364	1294 ± 410		0.877	49.5 (−678.0–777.0)	0.06		0.529	235.6 (−578.0–1049.0)	0.21		0.382	359.5 (−553.0–1272.0)	0.21
**Neurodegeneration**															
NfL (pg·ml^−1^)	5.3 ± 1.8	5.9 ± 2.0	4.2 ± 1.7		0.171	0.6 (−0.3–1.5)	0.45		<0.001	1.0 (−1.9–−0.1))	0.75		<0.001	1.6 (−2.3–−0.8)	1.43

Abbreviations: A, affected limb; NA, contralateral non-affected limb, NfL, neurofilaments; LEXT: leg extension exercise, PF, peak force; RFD, rate of force development; RPF, relative peak force; SQ, squat exercise; TUG, time up-and-go; VL, vastus lateralis; 60STS, 60 s sit-to-stand test.

## Data Availability

The data presented in this study are available on request from the corresponding author.

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
