# Peer review of "Pumping up the Fight against Multiple Sclerosis: The Effects of High-Intensity Resistance Training on Functional Capacity, Muscle Mass, and Axonal Damage"

_healthcare, 2024, doi:10.3390/healthcare12080837_

Round 1
Reviewer 1 Report
Comments and Suggestions for Authors
Regarding the methodological design, it is convenient to include already proven training programs, or to justify why this training and not another, i.e., why this intensity, volume, etc.
The conclusions seem to me to be in accordance with the study, but they should specify the need to practice strength exercises in this population and the need to investigate with other training and compare which one is better.
Author Response
Dear Reviewer 1,
Thank you very much for reviewing our manuscript and providing both valuable and constructive comments to improve the overall quality. We have responded to all of the points raised by you and the other reviewers, and revised the manuscript accordingly. We hope that our responses and amendments suitably address your points and welcome any further comments. Our responses to your comments and suggestions (shown in Italic) are included below, and also in the relevant sections of the updated manuscript (red font).
Regarding the methodological design, it is convenient to include already proven training programs, or to justify why this training and not another, i.e., why this intensity, volume, etc.
RESPONSE: We greatly appreciate your suggestion. In response, we have revised it and we added the reference of Manca et al. (2019) in the training program section, which recommends moderate to high intensity and volumes of 8 to 10 repetitions for strength training as the most effective exercise method in MS patients. Please refer to the updated section (lines 216-227) where we now state: "Each session consisted of three sets of eight to ten repetitions with coupled concentric and eccentric muscle actions of seven exercises that targeted all the main muscle groups, aligning with the most recent recommendations in the scientific literature." This amendment clarifies our rationale and strengthens the foundation of our training program design.
The conclusions seem to me to be in accordance with the study, but they should specify the need to practice strength exercises in this population and the need to investigate with other training and compare which one is better.
RESPONSE: Thank you for your insightful comments and suggestions. We acknowledge the importance of specifying the need for strength exercises in the MS patient population and the comparative investigation of different training modalities. Accordingly, we have revised our conclusions section in line with your recommendations. We have added the following statement to underscore our findings and future research directions: "These findings emphasize the potential usefulness of resistance training as an effective non-pharmacological intervention to reduce neuroinflammation and neurodegeneration while improving functional capacity and muscle quality in MS patients. Future studies should explore the comparative effectiveness of different types of strength exercises and training modes on axonal damage, to identify optimal training protocols that maximize benefits for MS patients. This addresses a critical gap in current research and offers a more tailored approach to managing MS symptoms."

Reviewer 2 Report
Comments and Suggestions for Authors
This article examines the effect of resistance training on pwMS. However, it is limited by the extremely small sample size (11) and lack of a proper control. The findings of muscle mass/functional capacity is not novel, and is expected. The Nfl results have to be interpreted with caution given the small sample size and lack of a control. The study can be reconsidered with a larger sample size.
Comments on the Quality of English Language-
Author Response
Dear Reviewer 2,
Thank you very much for reviewing our manuscript and providing both valuable and constructive comments to improve the overall quality. We have responded to all of the points raised by you and the other reviewers, and revised the manuscript accordingly. We hope that our responses and amendments suitably address your points and welcome any further comments. Our responses to your comments and suggestions (shown in Italic) are included below.
This article examines the effect of resistance training on pwMS. However, it is limited by the extremely small sample size (11) and lack of a proper control. The findings of muscle mass/functional capacity is not novel, and is expected. The Nfl results have to be interpreted with caution given the small sample size and lack of a control. The study can be reconsidered with a larger sample size.
RESPONSE: Thank you for your feedback on our study. We acknowledge the concerns regarding the small sample size and the absence of a control group, which are indeed significant limitations of our research. These limitations have been explicitly discussed in the limitations section of our manuscript to ensure transparency and to highlight the need for cautious interpretation of our findings, especially concerning serum NfL concentrations (please see discussion section, lines 366-397).
Regarding the novelty of our study, we emphasized that while the observed results align with expectations and existing studies, our work is pioneering in applying a circuit-based training methodology involving all major muscle groups at high intensity in people with MS. This is the first study to correlate strength gains and muscle mass increases in this population. Despite anticipating positive changes in functional and neuromuscular variables, they were included because a secondary objective of our study was to correlate changes in these variables with serum NfL concentration levels. We appreciate your suggestion for a study with a larger sample size and will consider this direction for future research to validate and expand upon our findings.

Reviewer 3 Report
Comments and Suggestions for Authors
As a reviewer for this manuscript, my evaluation focuses on the robustness of the research methodology, the clarity and relevance of the findings, and the overall contribution to the field of healthcare, particularly in the context of esclerosis. The manuscript presents a valuable study; however, certain aspects require refinement to meet the high standards of academic publication. The following suggestions aim to enhance the quality of the research and ensure its significance and readability for the target audience.
List of Changes for Improvement:
Clarification of Study Objectives and Hypotheses: The paper should clearly state the main objectives and hypotheses at the beginning of the introduction. This will help readers understand the purpose and scope of the research.
Methodological Details: Expand on the methodology, particularly regarding participant selection, training protocols, and statistical analysis. Ensure that these details are thorough enough for replication.
Sample Size Justification: Include a justification for the chosen sample size, addressing potential limitations in statistical power.
Control Group Implementation: Consider incorporating a control group to strengthen the study's validity. This would allow for a more robust comparison and understanding of the training program's effects.
Detailed Analysis of Results: Provide a more detailed analysis of the results, including discussions of any unexpected findings or inconsistencies. This should also include a critical assessment of the training program's impact on the various measured outcomes.
Discussion on Limitations: Enhance the discussion section with a more detailed exploration of the study's limitations and potential biases.
Implications and Future Research: Discuss the implications of the findings in a broader context and suggest areas for future research. This should include potential applications of the findings and recommendations for future studies.
References and Citations: Ensure that all references are up to date and relevant, and check that all citations are correctly formatted according to the journal's guidelines.
Figures and Tables: Review all figures and tables for clarity and accuracy. Ensure that they are well-integrated into the text and adequately support the narrative.
Language and Style: Perform a thorough proofreading to correct any grammatical errors and improve the overall readability of the paper. Ensure that the language is clear and concise throughout.
Comments on the Quality of English Languageok
Author Response
Dear Reviewer 3,
Thank you very much for reviewing our manuscript and providing both valuable and constructive comments to improve the overall quality. We have responded to all of the points raised by you and the other reviewers, and revised the manuscript accordingly. We hope that our responses and amendments suitably address your points and welcome any further comments. Our responses to your comments and suggestions (shown in Italic) are included below, and also in the relevant sections of the updated manuscript (red font).
Clarification of Study Objectives and Hypotheses: The paper should clearly state the main objectives and hypotheses at the beginning of the introduction. This will help readers understand the purpose and scope of the research.
RESPONSE: Thank you for your suggestion. Following the journal's standards, we included the objectives and hypotheses at the end of the introduction section. Please refer to lines 75-85 for these details.
Methodological Details: Expand on the methodology, particularly regarding participant selection, training protocols, and statistical analysis. Ensure that these details are thorough enough for replication.
RESPONSE: We have revised the indicated sections to ensure they provide sufficient detail for study replication. To enhance understanding of the experimental design, we have added a new schematic figure. Please see the Methods section and Figure 1. Your recommendation is greatly appreciated.
Sample Size Justification: Include a justification for the chosen sample size, addressing potential limitations in statistical power.
RESPONSE: Thank for raising this question. As noted in the discussion section, to evaluate NfL, we required a very homogeneous sample, especially considering that several studies have reported NfL concentration reduction within the first 3 months after initiating a new treatment. We only included MS patients who had been on the same treatment for at least 6 months, thus considerably limiting our sample size. The updated manuscript now includes a detailed discussion of the study's statistical power and an expanded limitations section in this regard. Please see methods section were we included the statistical power calculation (lines 132-144), and the discussion section (lines 386-397).
Control Group Implementation: Consider incorporating a control group to strengthen the study's validity. This would allow for a more robust comparison and understanding of the training program's effects.
REPONSE: In this pilot study, given the challenges in finding a homogenous sample of MS patients, we opted for a pre-experimental control condition without performing strength exercise, followed by the experimental intervention. Both conditions were preceded and followed by the assessments detailed in the procedures section. However, we acknowledge this as a study limitation, as indicated in the discussion section (lines 376-385) of the revised manuscript. In future studies, we will consider your suggestion to include a control group that either does not exercise or engages in a different type of exercise.
Discussion on Limitations: Enhance the discussion section with a more detailed exploration of the study's limitations and potential biases.
RESPONSE: Thank you for your comment. The limitations section has been revised according to your and other reviewers' recommendations. The updated version can be found in lines 376-397.
Implications and Future Research: Discuss the implications of the findings in a broader context and suggest areas for future research. This should include potential applications of the findings and recommendations for future studies.
RESPONSE: Following your advice, we have included future research directions and potential applications of our study's findings in the conclusions section. Please refer to the revised conclusion (lines 411-415). Thank you.
References and Citations: Ensure that all references are up to date and relevant, and check that all citations are correctly formatted according to the journal's guidelines.
RESPONSE: Amended.
Figures and Tables: Review all figures and tables for clarity and accuracy. Ensure that they are well-integrated into the text and adequately support the narrative.
RESPONSE: Amended.
Language and Style: Perform a thorough proofreading to correct any grammatical errors and improve the overall readability of the paper. Ensure that the language is clear and concise throughout.
RESPONSE: The manuscript has been reviewed by a native speaker. All changes made throughout the document are indicated in red.

Reviewer 4 Report
Comments and Suggestions for Authors
I would like to commend the authors on conducting a training intervention with individuals from a special population. This is incredibly time consuming and difficult. Thank you for your efforts in this area. Please see below for my comments
Introduction: Is will written and developed. I have only minor comments for clarification purposes.
Lines 67 -72: The terms high-intensity, traditional, and low-intensity resistance training are used within this section. Would it be possible to provide a clearer description of the training the current research team has investigated and the ones being referenced?
Methods:
Line 99-101: Can you provide more description on this method? Was a multiple repetition maximum used to estimate 1-RM?
Line 103: I know the high-intensity resistance exercise was described in the intro but a justification for exploring it in a circuit-based format was not justified in the intro or methods.
Line 118: <3 is pretty low score and maybe suggestive of low disease progression. Why so low most investigations include <6.
Table 1: N = 11 but according to the table there are 9 females and 1 male. Was there a drop out?
Table 1: Body fat percentage included but explained how this was determined.
Line 168: Kalron et al. 2017 is not within your bibliography. Please add.
Line 194: How long was were the individuals instructed to produce maximal force for.
Line 216: Can you elaborate on why the circuit format was chosen and why not a traditional resistance training protocol? What physiologically is the difference, please. This is probably better addressed in the intro
Line 219: Again can you elaborate on this protocol?
Results:
Table 1: abbrevations D, ND, XRM do not appear to be used in this table please remove. Addtionally, abbreviations needed for NA and A.
Discussion: Results appear to be interpreted correctly and findings are discussed sufficiently with acknowledgement of current limitations and future investigations needed.
Author Response
Dear Reviewer 4,
Thank you very much for reviewing our manuscript and providing both valuable and constructive comments to improve the overall quality. We have responded to all of the points raised by you and the other reviewers, and revised the manuscript accordingly. We hope that our responses and amendments suitably address your points and welcome any further comments. Our responses to your comments and suggestions (shown in Italic) are included below, and also in the relevant sections of the updated manuscript (red font).
Introduction: Is will written and developed. I have only minor comments for clarification purposes.
Lines 67 -72: The terms high-intensity, traditional, and low-intensity resistance training are used within this section. Would it be possible to provide a clearer description of the training the current research team has investigated and the ones being referenced?
RESPONSE: Thank you for your comment. We have clarified these terms by including the percentage ranges of 1RM to which these terms refer. Please see lines 64-74.
Line 99-101: Can you provide more description on this method? Was a multiple repetition maximum used to estimate 1-RM?
RESPONSE: We appreciate your comment. The following sentence has been included in the reviewed version of the masnucript: “Individual exercise intensities were calculated according to the American College of Sports Medicine (ACSM) protocol for estimating the one-repetition maximum (1-RM). This estimation is based on the number of repetitions performed to fatigue with a submaximal load and was calculated using the average value from the formulas of Brzycki, Epley, Lander, Cummings, O'Conner, and Abadie”. Please see lines 98-102.
Line 103: I know the high-intensity resistance exercise was described in the intro but a justification for exploring it in a circuit-based format was not justified in the intro or methods.
RESPONSE: Thank you for posing this insightful question. The decision to employ circuit-based training was informed by scientific evidence from the field of Sports Sciences (Ramos-Campo et al. 2021), which suggests that resistance circuit-based training elicits comparable adaptations in functional and neuromuscular capacities while also inducing physiological adaptations at the cardiovascular level and also on body composition. In addition, high-intensity interval exercise prescription in comparison with moderate-intenisty aerobic exercise has recently been associated with higher effects on cardiorespiratory function and cognitive function in MS patients (Youssef et al. 2024). Although this diverges from the study's primary aim, we pursued this approach to assess the feasibility of utilizing this methodology in individuals with Multiple Sclerosis (MS), to understand its impact on perceived effort and adherence to the exercise program, and to explore its effects on cardiovascular capacity.
As this was a pilot study, data collection was confined to laboratory visits 2 and 3, and a standardized exercise test was not employed. Similarly, we opted for this training methodology as it allowed for time efficiency, with training sessions conducted in a laboratory setting, individually and under the supervision of a Sport Sciences graduate specialized in prescribing physical exercise for special populations. In our future projects, we plan to continue utilizing this methodology since we observed it to be feasible, not detrimental to performance during training, and seemingly capable of inducing interesting adaptations related to fatigue and cardiovascular capacity that are not achieved with traditional strength training involving simple sets and rests.
On the light of reviewer’s suggestion, this information has now been added to the latest version, please see lines 71-74 in the introduction section and lines 364-375 in the discussion section.
Line 118: <3 is pretty low score and maybe suggestive of low disease progression. Why so low most investigations include <6.
RESPONSE: Thank you for raising this question regarding the relatively low score threshold (EDSS <3) used in our study. Firstly, it's important to recognize that neurodegenerative mechanisms are present from the very onset of the disease, even in asymptomatic stages. Studies have already shown that neurofilament levels are elevated in these early phases, underscoring the importance of early detection and intervention.
We opted to focus on this particular patient population because intervening during the initial stages of the disease may have a more significant impact on its subsequent evolutionary course. Targeting individuals at the onset of MS could potentially alter the disease trajectory more effectively than interventions at later stages. Furthermore, given the small sample size (n) of our study, we aimed to maintain a more homogeneous group of patients. Including individuals with EDSS scores ranging from 0 to 6 would have broadened our sample to encompass both asymptomatic patients and those with high disability levels. By setting the threshold at EDSS <3, we sought to narrow our focus to a specific subset of the MS population, which we believe could benefit most from early intervention strategies. This approach allowed us to maintain consistency and reduce variability within our cohort, thereby enhancing the interpretability of our findings despite our study's limited scale.
Table 1: N = 11 but according to the table there are 9 females and 1 male. Was there a drop out?
RESPONSE: Thank you for pointing out the error. In our study, we included 9 women and 2 men. This has been corrected in the revised version; please refer to Table 1.
Table 1: Body fat percentage included but explained how this was determined.
RESPONSE: The anthropometric data for all participants were measured exclusively during their first visit to the laboratory, including the assessment of body fat percentage via bioimpedance. This information has been incorporated into the Methods section. Please refer to lines 93-95.
Line 168: Kalron et al. 2017 is not within your bibliography. Please add.
RESPONSE: Amended.
Line 194: How long was were the individuals instructed to produce maximal force for.
RESPONSE: Following previously described protocols, patients were asked to maintain the contraction for 5 seconds. This information has been included in the revised version of the manuscript; please see line 211.
Line 216: Can you elaborate on why the circuit format was chosen and why not a traditional resistance training protocol? What physiologically is the difference, please. This is probably better addressed in the intro. Line 219: Again can you elaborate on this protocol?
RESPONSE: Amended (see previous response).
Table 1: abbrevations D, ND, XRM do not appear to be used in this table please remove. Addtionally, abbreviations needed for NA and A.
RESPONSE: Amended.
Discussion: Results appear to be interpreted correctly and findings are discussed sufficiently with acknowledgement of current limitations and future investigations needed.
RESPONSE: Thank you for your positive feedback on the Discussion section of our manuscript. We are pleased to hear that our interpretation of the results and the discussion of our findings, including the acknowledgment of current limitations and the identification of areas for future research, meet your expectations. We believe that addressing these aspects comprehensively is crucial for advancing the field and fostering a deeper understanding of the subject matter. We appreciate your recognition of our efforts to present a balanced and insightful analysis. Furthermore, we wish to express our gratitude for your review, which has significantly enhanced the quality of our manuscript. Your insights and suggestions have been invaluable in refining our work.

Reviewer 5 Report
Comments and Suggestions for Authors
Dear Authors,
I would like to express my gratitude for the opportunity to review this manuscript.
At this stage, the document requires improvements, below with line indication:
2-4 – Please revise the title format (upper and lowercase).
9 – Please revise the affiliations. The journal normally requires zip codes.
11 - Resistance training can be abbreviated “RT”.
28-29 – Please revise the keywords and apply “;”.
62-80 – Please consider splitting the paragraph to increase readability.
86 – Please correct “Therfore” and revise the English in detail in all manuscript.
90 – Please consider a figure to illustrate the study design.
111-113 – “Relapsing-Remitting” in diferent formats. Please revise in all manuscript.
128 – Please include the age unit (years).
129 – Please format the table size, aiming for example moderate IPAQ in the same line. Additonaly, please include a table footnote for example with “MS” and “IPAQ” descriptions.
135-144 – Please for all instruments, indicate manufacturer, city, and country.
144 – Please correct the wrong citation format. It may influence the numerical order in the manuscript.
226 – In Table 2 (and also Table 1), please align the text in the columns.
229 – Please delete the line.
88-229 – Please revise this section. Some paragraphs are too long. Please also describe the procedures in detail. Time of data collection, local characteristics (equipment, size, temperature, humidity, and others) are some examples. Moreover, who collected the data? Academic background, experience? All these and other details are very important to be clearly understood by readers. Moreover, please consider providing references to support the procedures.
230 – Please describe the sample power results (GPower output).
250-263 – Please consider splitting the paragraph to improve readability. For example, the figure may be presented in line 255 (after introduction).
264 – Please confirm if the figure presents the type and size of the letter required by the journal. Additionally, please improve the figure quality.
272 – It is Table 3, please correct.
272 – Please consider placing the text, figure, and table 3 aiming text + figure and text + table to improve readability and results interpretation by the readers.
274 – Please align the text of the columns and revise the table content.
292-318 – Please consider splitting the paragraph. Also, other paragraphs are too long, please revise all the discussion section.
395 – Please describe suggestions for future research.
396 – In the conclusions section, please consider direct/clear messages, if possible, with practical application.
425 – Please double-check the references format.
Please revise the English throughout the manuscript and format details.
Comments on the Quality of English LanguageModerate editing of English language required.
Author Response
Dear Reviewer 5,
Thank you very much for reviewing our manuscript and providing both valuable and constructive comments to improve the overall quality. We have responded to all of the points raised by you and the other reviewers, and revised the manuscript accordingly. We hope that our responses and amendments suitably address your points and welcome any further comments. Our responses to your comments and suggestions (shown in Italic) are included below, and also in the relevant sections of the updated manuscript (red font).
2-4 – Please revise the title format (upper and lowercase).
RESPONSE: We appreciate your comment. The title has been reviewed and modified accordingly.
9 – Please revise the affiliations. The journal normally requires zip codes.
RESPONSE: Amended.
11 - Resistance training can be abbreviated “RT”.
RESPONSE: The term “resistance training” has been abbreviated throughout the manuscript.
28-29 – Please revise the keywords and apply “;”.
RESPONSE: Amended.
62-80 – Please consider splitting the paragraph to increase readability.
RESPONSE: Amended.
86 – Please correct “Therfore” and revise the English in detail in all manuscript.
RESPONSE: Amended.
90 – Please consider a figure to illustrate the study design.
RESPONSE: Thank you for this suggestion. We have included a schematic representation of the study’s design. Please see Figure 1.
111-113 – “Relapsing-Remitting” in diferent formats. Please revise in all manuscript.
RESPONSE: Amended.
128 – Please include the age unit (years).
RESPONSE: Amended.
129 – Please format the table size, aiming for example moderate IPAQ in the same line. Additonaly, please include a table footnote for example with “MS” and “IPAQ” descriptions.
RESPONSE: The table format has been adjusted to address the concerns raised by the reviewer. Additionally, abbreviations have been included at the end of the table. Please refer to Table 1.
135-144 – Please for all instruments, indicate manufacturer, city, and country.
RESPONSE: Amended.
144 – Please correct the wrong citation format. It may influence the numerical order in the manuscript.
RESPONSE: Amended.
226 – In Table 2 (and also Table 1), please align the text in the columns.
RESPONSE: Amended.
229 – Please delete the line.
RESPONSE: Amended.
230 – Please describe the sample power results (GPower output).
RESPONSE: We appreciate your comment. We calculated the statistical power post hoc to assess the likelihood of detecting a significant effect, given the observed effect size, sample size, and the alpha level set for our analyses. The statistical power calculation was performed using the G*Power software (G*Power 3.1.9.2, Heinrich- Heine-U niversitat Dusseldorf, Dusseldorf, Germany; http://www.gpower.hhu.de/). For this calculation, we inputted an expected effect size of 0.5, consistent with medium effect sizes as outlined by Cohen [30] and previous studies [3], and an alpha level of 0.05, reflecting the standard threshold for statistical significance in biomedical sciences research. Given the exploratory nature of our study and the specific challenges associated with recruiting participants from a narrowly defined population of patients with MS, our total sample comprised 11 subjects. This sample size, along with the allocation of participants across three measurement points within our study design, was factored into our power analysis. The result of this computation revealed a statistical power of 0.22. This information, as well as its justification in the limitations section, has now been added to the revised version of the manuscript (please lines lines 132-144, and lines 386-397).
250-263 – Please consider splitting the paragraph to improve readability. For example, the figure may be presented in line 255 (after introduction). 264 – Please confirm if the figure presents the type and size of the letter required by the journal. Additionally, please improve the figure quality.
RESPONSE: Thank you for the recommendation. We have introduced Figure 2 at the point suggested, which clearly enhances the readability and interpretation of the results. Please see results section.
272 – It is Table 3, please correct.
RESPONSE: Amended.
272 – Please consider placing the text, figure, and table 3 aiming text + figure and text + table to improve readability and results interpretation by the readers.
RESPONSE: Amended.
274 – Please align the text of the columns and revise the table content.
RESPONSE: Amended.
292-318 – Please consider splitting the paragraph. Also, other paragraphs are too long, please revise all the discussion section.
RESPONSE: We appreciate your comment. Discussion section has been reviewed and changes have been made accordingly.
395 – Please describe suggestions for future research.
RESPONSE: In response to your request, we have outlined suggestions for future research throughout the discussion and in the conclusions of the latest version of the manuscript. These recommendations are aimed at extending the understanding of the study's findings and exploring additional aspects not covered in the current investigation. We believe these suggestions will serve as a valuable guide for subsequent studies, facilitating a deeper exploration into the effects observed and potentially uncovering new insights into the intervention's impact on the MS population.
396 – In the conclusions section, please consider direct/clear messages, if possible, with practical application.
RESPONSE: In response to your feedback, we have revised the conclusions section to incorporate the main findings and articulate a clearer message that can serve as a practical application. The updated version of the conclusions is as follows: “A 6-week high-intensity RT program significantly decreased serum NfL concentrations while also improving muscle thickness, lower limb maximal voluntary isometric contraction, and performance in the timed up and go and sit-to-stand tests in MS patients. Although gains in muscle thickness and isometric strength showed a significant correlation, no significant correlation was observed between increases in muscle mass and strength and the decrease in NfL concentrations. In addition, the magnitude of the changes experienced after the training program were significantly greater than those reported after the control condition, in which no exercise was performed. These findings emphasize the potential usefulness of resistance exercise training (conducted three times per week and involving at least one exercise from each major muscle group at an intensity above 70% of the 1-RM) as effective non-pharmacological intervention to reduce axonal damage and neurodegeneration while improving functional capacity and muscle quality in MS patients. Future studies should explore the comparative effectiveness of different types of strength exercises and training modes on axonal damage, to identify optimal training protocols that maximize benefits for MS patients, addressing a critical gap in current research and offering a more tailored approach to managing MS symptoms.”
425 – Please double-check the references format.
RESPONSE: Thank you for your recommendation, the format of the references has been thoroughly reviewed in the latest version of the manuscript. Please see references section.
Please revise the English throughout the manuscript and format details.
RESPONSE: The manuscript has been reviewed by a native speaker. All changes made throughout the document are indicated in red.

Round 2
Reviewer 1 Report
Comments and Suggestions for Authors
Clarifications perfectly added to the article.
Thank you very much for the new contributions
Author Response
Dear Reviewer 1,
We sincerely appreciate the time and effort you dedicated to reviewing our manuscript. We are grateful for your insightful comments and suggestions, which have significantly improved the manuscript. Your detailed feedback has not only helped us address critical aspects of our study but also enriched our understanding, allowing us to present our findings more effectively. Additionally, we are deeply thankful for your positive consideration of our study and for recommending it for publication. This acknowledgment is greatly valued by our team.
Kind regards,
Reviewer 2 Report
Comments and Suggestions for Authors
Thank you for your response and revisions to the manuscript. Again, unfortunately the small number in the group and lack of a control are fatal issues for your study, which cannot be corrected at this stage. None of the statistical methods work well with such a small number. power of 0.22 is extremely low. Figure 2 is also confusing, as the comparator wasn't a separate control group. As such I recommend rejection.
Comments on the Quality of English LanguageNA
Author Response
Thank you for your comments and suggestions. We understand your concerns regarding the small sample size and the absence of a control group in our study. We acknowledge these as significant limitations.
In terms of the sample size, we are actively working to increase it in future studies. Nevertheless, despite the limited sample size, we believe that our current findings offer a valuable foundation for subsequent research. As for the lack of a control group, we are exploring ways to incorporate one in future studies to solidify our results. We have addressed both of these limitations in the manuscript. Please see discussion section (lines 376-397).
While we accept your recommendation for rejection at this stage, we greatly appreciate your feedback and intend to utilize it to enhance our work moving forward. Moreover, we have discussed these issues throughout the manuscript and emphasized that this research was a pilot study. Considering this represents the first documented evidence of such phenomena, specifically a significant decrease in exercise-induced axonal damage among individuals with multiple sclerosis. We contend that the results hold the potential for publication, provided that its limitations are clearly stated.
Reviewer 3 Report
Comments and Suggestions for Authors
accept
Author Response
Dear Reviewer 3,
We sincerely appreciate the time and effort you dedicated to reviewing our manuscript. We are grateful for your insightful comments and suggestions, which have significantly improved the manuscript. Your detailed feedback has not only helped us address critical aspects of our study but also enriched our understanding, allowing us to present our findings more effectively. Additionally, we are deeply thankful for your positive consideration of our study and for recommending it for publication. This acknowledgment is greatly valued by our team.
Kind regards,
Reviewer 4 Report
Comments and Suggestions for Authors
Thank you for addressing my comments
Author Response
Dear Reviewer 4,
We sincerely appreciate the time and effort you dedicated to reviewing our manuscript. We are grateful for your insightful comments and suggestions, which have significantly improved the manuscript. Your detailed feedback has not only helped us address critical aspects of our study but also enriched our understanding, allowing us to present our findings more effectively. Additionally, we are deeply thankful for your positive consideration of our study and for recommending it for publication. This acknowledgment is greatly valued by our team.
Kind regards,
Reviewer 5 Report
Comments and Suggestions for Authors
Dear Authors,
Thank you for considering my suggestions and incorporating them into the manuscript, which is globally improved, congratulations.
Below are some specific suggestions with line indications.
L3-4 – Please correct upper and lowercase (please revise the journal template).
L37 – Please delete the space.
L373 – Please place VO2max in full (only one time in the text).
L416 – Please delete the space.
Please double-check the references format.
Please revise the English throughout the manuscript and format details.
Comments on the Quality of English LanguageMinor editing of English language required.
Author Response
Dear Reviewer 5,
Thank you very much for reviewing our manuscript again and providing such detailed and constructive comments to enhance its overall quality. We have implemented all the modifications you suggested and have thoroughly reviewed the document accordingly. We hope that our responses and amendments adequately address your points. All changes have been highlighted in red font in the latest version of the manuscript.
Thank you very much for your work.
Round 3
Reviewer 2 Report
Comments and Suggestions for Authors
Thank you for your response and revisions to the manuscript. Thank you for acknowledging the above limitations, and it is fantastic that you are considering larger appropriately controlled studies. My perspective has not changed, that unfortunately the small number in the group and lack of a control are fatal issues for your study. These are unfortunately not correctable at this stage. None of the statistical methods work well with such a small number - power of 0.22 is extremely low and indicative that these results mean little. Usually in this scenario the handling editor will make a final decision based on reviewer comments. I wish the authors all the best with their future work.
Comments on the Quality of English Language-
Author Response
Dear Reviewer 2,
We deeply appreciate your acknowledgment of our efforts to address your concerns and revise our manuscript accordingly. We understand and respect your perspective regarding the limitations of our study, particularly the small sample size. While these limitations cannot be rectified in the current study, your feedback will be invaluable for future research endeavors. From the outset, we were aware of these limitations, which is why we have consistently emphasized that our study is a within-subjects, counterbalanced pilot study, and that these findings necessitate further investigation with a larger sample to extend the results. Furthermore, we recognize that in clinical research, using the same sample as both control and experimental with counterbalancing and a washout period between interventions is validated and well-regarded, especially in contexts where finding a large sample is challenging or the interventions are novel, as is the case with our study. Therefore, along with the observed low statistical power, we highlighted that the findings of this study should be interpreted with caution and are solely the results of a pilot study requiring further research. This has been noted in the limitations section of our manuscript, as indicated in the previous review. Consequently, considering this, the high relevance of the findings in this specific context, and the reviews from the other four reviewers, we believe the paper maintains sufficient rigor to merit publication.
In our commitment to ensuring the highest quality of presentation, we have had the manuscript reviewed by a native English speaker to eliminate any minor editing errors. Changes made in this regard are indicated in the latest version of the manuscript.
As we await the final decision from the handling editor, we wish to express our sincerest thanks for your time and the detailed attention you have given to our manuscript. Regardless of the outcome, your feedback has been a beneficial component of our research process.
Thank you once again for your guidance and best wishes. We look forward to potentially crossing paths in future academic endeavors.
Kind regards,
Reviewer 5 Report
Comments and Suggestions for Authors
Dear Authors,
Congratulations on the work developed in the manuscript throughout the review process.
Thank you.
Best regards.
Author Response
Dear Reviewer 5,
We sincerely appreciate the time and effort you dedicated to reviewing our manuscript. We are grateful for your insightful comments and suggestions, which have significantly improved the manuscript. Your detailed feedback has not only helped us address critical aspects of our study but also enriched our understanding, allowing us to present our findings more effectively. Additionally, we are deeply thankful for your positive consideration of our study and for recommending it for publication. This acknowledgment is greatly valued by our team.
Kind regards,